

# Low-pressure gas chromatography with chemical ionization mass spectrometry for quantification of multifunctional organic compounds in the atmosphere

Krystal T. Vasquez[1], Hannah M. Allen[1], John D. Crounse[2], Eric Praske[1], Lu Xu[2], Anke C. Noelscher[2*], and Paul O. Wennberg[2,3]

[1]Division of Chemistry and Chemical Engineering, California Institute of Technology, Pasadena, CA 91125, USA
[2]Division of Geological and Planetary Sciences, California Institute of Technology, Pasadena, CA 91125, USA
[3]Division of Engineering and Applied Science, California Institute of Technology, Pasadena, CA 91125, USA
[*]now at: Deutscher Wetterdienst, Technische Infrastruktur, Frankfurter Straße 135, Offenbach am Main, Germany

*Correspondence to:* Krystal T. Vasquez (kvasquez@caltech.edu) and Paul. O. Wennberg (wennberg@caltech.edu)

**Abstract.** Oxygenated volatile organic compounds (OVOCs) are formed during the oxidation of gas phase hydrocarbons in the atmosphere. However, analytical challenges have hampered ambient measurements for many of these species, leaving unanswered questions regarding their atmospheric fate. We present the development of an *in situ* gas chromatography (GC) technique that, when combined with the sensitive and specific detection of chemical ionization mass spectrometry (CIMS),

5    is capable of the isomer-resolved detection of a wide range of OVOCs by addressing several issues typically associated with chromatographic separation of such compounds (e.g., analyte degradation). The performance of this instrumentation is assessed through data obtained in the laboratory and during two field studies. We show that this instrument is able to successfully measure otherwise difficult-to-quantify compounds (e.g., organic hydroperoxides and organic nitrates) and observe the diurnal variations of a number of their isomers.

## 10   1   Introduction

The composition of the atmosphere is determined through a dynamic array of chemical emission, transport, deposition and photochemical processing. Our ability to accurately predict future trends of both air quality and climate change depends on understanding these processes. Of particular interest is the photooxidation of non-methane hydrocarbons (NMHCs) that, due to their high abundance, influence the distributions of key atmospheric constituents such as ozone ($O_3$) and secondary organic

15    aerosol (SOA). While decades of research have provided much insight into the link between atmospheric composition and chemistry, significant knowledge gaps still persist and the atmospheric degradation pathways of many NMHCs remain poorly understood.

    The gas phase oxidation of NMHCs is typically initiated by one of several atmospheric oxidants (e.g., OH, $NO_3$, or $O_3$) converting these hydrocarbons into oxygen-containing, often multifunctional, intermediates. These first-generation oxygenated

20    volatile organic compounds, or OVOCs, can undergo further transformations through a number of competing physical and photochemical sinks (Atkinson and Arey, 2003; Mellouki et al., 2015), each of which can have a unique effect on the atmosphere.





Some OVOCs can undergo photochemical fragmentation to smaller species, often through conversion of NO to $NO_2$ leading to local ozone formation, while others (such as those with longer atmospheric lifetimes) can be transported downwind prior to oxidation, extending their effects to regional and global scales. Chemical oxidation can also lead to a scenario in which the OVOCs increase their functionality creating large, low-volatility, multifunctional products that partition into the particle

phase and contribute to the formation and growth of aerosol. In addition, it has also been shown that significant portions of OVOCs can be removed from the atmosphere through fast deposition processes (Nguyen et al., 2015) which can greatly affect the chemical cycling of many important compounds.

It is the relative importance of each possible sink that establishes the dominant tropospheric fate of these compounds and thereby the impact of their hydrocarbon precursors (Koppmann and Wildt, 2008). This seemingly straightforward relationship

can quickly become complicated however, especially for larger compounds ($>C_3$). A prime example of this can be seen during the OH oxidation of isoprene, a highly abundant and reactive biogenic VOC, which produces six isomeric peroxy radicals ($RO_2$). Changes in the relative abundance of these radicals can result in vastly different ratios of its OVOC products (Orlando and Tyndall, 2012; Teng et al., 2017; Wennberg et al., 2018), allowing isoprene to either have a profound effect on ozone and SOA through its bimolecular reaction products—isoprene hydroxy nitrates (IHN) and isoprene hydroxy hydroperoxides

(ISOPOOH), respectively—or on the OH radical which can be recycled during the subsequent chemistry of products that arise from the unimolecular $RO_2$ reaction channel (e.g. hydroperoxy aldehydes or HPALDs; Peeters et al., 2014). These structural effects are also apparent throughout the later generation chemistry of isoprene and other NMHCs and the outputs of global chemistry transport models can be quite sensitive to this isomer-specific chemistry. For example, ozone production, in particular, has been shown to be highly dependent on the assumed yields and reaction rates of specific organic nitrate isomers

(Squire et al., 2015), which together determine the net $NO_x$ recycling capabilities of each compound.

Despite its importance, our understanding of this intricate chemistry has been hindered by the lack of instrumentation capable of providing isomer-resolved measurements of important OVOCs. Recent progress has been made in this respect for laboratory studies (e.g., Bates et al., 2014, 2016; Lee et al., 2014; Teng et al., 2015, 2017; Schwantes et al., 2015; Praske et al., 2015, 2018). Analytical techniques for ambient measurements, however, either suffer from high detection limits and/or large instrumental

losses of these reactive analytes (Vairavamurthy et al., 1992; Apel et al., 2003, 2008; Clemitshaw, 2004), and so the focus has been typically on smaller, more abundant compounds (Mellouki et al., 2003; Koppmann and Wildt, 2008; Hellén et al., 2017).

Gas chromatography (GC) can reach the detection limits needed to measure a variety of larger OVOCs by preconcentrating analytes prior to separation and utilizing detection methods such as flame ionization detection (FID) or electron impact mass spectrometry (EI-MS) (Ras et al., 2009). As a result, this technique is becoming increasingly popular and has recently been

used for the *in situ* detection of carbonyls (Apel et al., 2003), organic acids (Hellén et al., 2017), organic nitrates (Mills et al., 2016) and other oxygenated organic compounds (e.g., Clemitshaw, 2004; Koppmann and Wildt, 2008; Roukos et al., 2009). Nevertheless, these GC techniques come with their own analytical challenges as the non-specificity of GC-FID and overall difficulty in differentiating fragmentation patterns of isobaric and isomeric species can create data sets that hide the intricacies of crucial structure-activity relationships of individual compounds. In addition, the multifunctional nature of these compounds

makes them highly reactive, increasing the likelihood that they will be lost or converted into different species through surface-





enhanced reactions that can occur at various stages of GC analysis. Converted species can be subsequently detected (e.g., Rivera-Rios et al., 2014), thus identifying such artifacts necessitates authentic calibrations even for species not being targeted. Due to the lack of commercially available standards for many species of interest, this can quickly become labor intensive or simply not feasible, leading to large uncertainties in these types of measurements and much confusion regarding chemical

mechanism elucidation.

Here, we present the development and deployment of a new gas chromatography method that uses the highly sensitive detection of chemical ionization mass spectrometry (CIMS) for the near real-time detection of a number of OVOCs. With this instrumentation, we address many of the historical issues associated with the use of gas chromatography for atmospheric field sampling, allowing for the preservation of difficult-to-measure compounds and enabling isomer-resolved measurements of a

wide array of compounds. Compounds discussed in this study are shown in Table 1. To distinguish between different isomers of the hydroxynitrates, ISOPOOH, HPALD and ICN, we employ an abbreviated naming scheme in which the first number denotes the carbon position where the oxidant originally adds to the parent alkene and the second denotes the position of the additional functional group (e.g. for 1,2-IHN the hydroxy group added to the C1 carbon of isoprene, followed by a nitroxy group at C2).

## 2   Instrument Description

A simplified schematic of the GC-HR-ToF-CIMS is shown in Fig. 1. It integrates the use of a metal-free, low-pressure gas chromatograph (LP-GC) positioned upstream of a high-resolution time-of-flight chemical ionization mass spectrometer (HR-ToF-CIMS, TofWerk/Caltech), allowing for two sampling modes: (1) direct atmospheric sampling for the real-time quantification of gas-phase species (hereafter, direct CIMS sampling), and (2) GC-CIMS analysis for the collection, separation and

quantification of ambient isomer distributions of select OVOCs. The overall design of this instrumentation is based upon an existing testbed that has been used in previous laboratory studies (e.g., Bates et al., 2014; Lee et al., 2014; Teng et al., 2015, 2017; Schwantes et al., 2015). Here, we have automated and field-hardened this design such that its novelty comes from the capability to operate under a variety of field conditions with minimal maintenance as it captures real-time data through a programmed sampling routine.

### 2.1   HR-ToF-CIMS

The HR-ToF-CIMS builds upon methods developed with a previous custom-built quadropole CIMS (Crounse et al., 2006, later upgraded to a c-ToF-CIMS). Ambient air is drawn at high flow rate (~2000 $\mathrm{slm}$, P ~1 $\mathrm{atm}$) through a custom Teflon-coated glass inlet (3.81 $\mathrm{cm}$ I.D x 76.2 $\mathrm{cm}$ long) after which a small fraction of the flow is sub-sampled perpendicular to the main flow in order to discriminate against large particles that may be present. This sub-sampled gas stream can be directed to the

CIMS, the GC, or a zeroing system through short lengths of 6.35 $\mathrm{mm}$ O.D. PFA tubing. When measured directly by the CIMS, the sample first flows through a fluoropolymer-coated (Cytonix PFC801A) glass flow tube (Fig. 1F) maintained at 35 $\mathrm{mbar}$ before undergoing chemical ionization by a $\mathrm{CF_3O^-}$ reagent ion (*m/z* 85) whose chemistry has been described in more detail





elsewhere (Huey et al., 1996; Amelynck et al., 2000a, b; Crounse et al., 2006; Paulot et al., 2009a, b; St. Clair et al., 2010; Hyttinen et al., 2018). Briefly, $CF_3O^-$ is formed by passing 380 sccm of 1 ppmv $CF_3OOCF_3$ in $N_2$ through a cylindrical tube (Fig. 1G) containing a layer of polonium-210 (NRD LLC Po-2021, initial activity: 10 mCi). Alpha-particles produced from the radioactive decay of the polonium react with the $N_2$ gas to produce electrons which react rapidly with $CF_3OOCF_3$ to produce

the $CF_3O^-$ ion. The reagent ion interacts with the analytes by forming cluster ($m/z$ = analyte mass + 85) or fluoride transfer ($m/z$ = analyte mass + 19) product ions allowing for the detection of small organic acids and other oxygenated multifunctional compounds with high sensitivity (LOD ≈ 10 pptv for 1 s integration period) and minimal fragmentation.

Following ionization, the ions are directed via a conical hexapole ion guide into the high resolution mass spectrometer (Tofwerk) which collects data for masses between $m/z$ 19 and $m/z$ 396 at 10 Hz time resolution. The HR-ToF CIMS has a mass

resolving power of ~3000 m/dm, allowing for the separation of some ions with different elemental composition but the same nominal mass.

## 2.2   GC

Chromatographic separation of analytes is achieved on a short (1-m) column encased between two aluminum plates, each measuring 130 mm x 130 mm x 5 mm (total mass = 466 g , Fig. 2). The column sits within a rectangular groove (0.8 mm wide

x 2.4 mm deep) machined into one plate, which serves to both hold the column in place and allow for it to make good thermal contact with the metal as it makes 2.5 loops around the plate. The temperature of the GC assembly can be controlled over a large range, cooling to -60°C using liquid $CO_2$ and warming to ~200°C, reaching a maximum heating rate of 42°C min$^{-1}$ with its electrical heating system (described in Sect. 2.2.1). In addition, the entire GC system is automated and the majority of its processes operate in parallel with direct CIMS sampling to allow for minimal interruptions in instrument sampling. The GC

system is also modularized, containing its own control system, enabling its use with other detectors.

### 2.2.1   Operating Parameters

For the studies detailed in this paper, air is subsampled from the main instrument inlet and directed into the cryocooled 0.53 mm I.D. RTX-1701 megabore column (Restek) at a flow rate of 220 sccm. Ambient air is diluted by a factor of 15 to 30, depending on the relative humidity (RH) of the sample, and the targeted compounds are collected over a 10-minute period on

the column head at -20°C. As discussed in later sections, the choice of the dilution and trapping temperature is a compromise between adequately cryofocusing the maximum amount of analytes while avoiding the collection of water. After collection, a Teflon solenoid valve (SH360T042, NResearch) is switched allowing $N_2$ carrier gas to enter the column at a constant flow rate of 5 sccm (Horiba ZS12, Fig. 1N). The compounds are separated using a programmable temperature controller (Watlow F4 series) and several resistance heaters (~400 total watts, KH series, Omega) adhered to the outside of each plate. The automated

temperature program proceeded as follows: a 3 minute temperature ramp to 20°C (~13°C min$^{-1}$), followed by a 3°C min$^{-1}$ ramp to 50°C, followed by a 10°C min$^{-1}$ increase to 120°C for a total temperature ramping time of 20 minutes. Following completion of the temperature program, the column is held at 120°C for an additional two minutes to remove remaining analytes.





### 2.2.2 GC Cooling System

The GC assembly is cooled through the expansion of liquid $CO_2$ entering from the center of each plate. The $CO_2$ flows along eight radial grooves. An o-ring seal contains the $CO_2$ and causes it to exit via ports machined into the plate near the radius of the column. To achieve sufficient time resolution for the GC measurements (1 cycle per hour), the column must cool to the

cryotrapping set point within a short time period regardless of ambient temperatures. However, we also wish to minimize the $CO_2$ usage, reducing the maintenance required in the field. Thus $CO_2$ flow is controlled into the GC plates using two solenoid valves (Series 9, Parker) connected to ~29 cm x 0.25 mm ID and ~35 cm x 0.15 mm I.D. PEEK restrictors. With both valves open, a total $CO_2$ flow rate of 25 slm (as gas) is admitted to cool the GC assembly from 67°C to -20°C within the allotted 10 minute period. To conserve $CO_2$ while maintianing the trapping temperature, only a single $CO_2$ valve is opened (see below).

### 2.2.3 Cyrotrap Temperature Control

During the collection of analytes on the head of the column, it is important that the temperature remains stable, as sizable fluctuations in temperature adversely affects the chromatography. To control the trapping set point, we utilize the heaters and the resistance temperature detector (RTD, F3102, Omega) located on the GC column ring (Fig. 2, #2 on diagram) in a PID control loop. In addition, during trapping we only use the solenoid valve connected to the 0.15 mm I.D. restrictor as this valve

provides a $CO_2$ flow that is adequate to maintain the GC temperature (~10 slm).

Additional efficiency was gained by insulating the GC assembly with Nomex™ felt and wrapping the felt with Kapton tape to prevent water vapor from diffusing to and condensing on the cold plates. The entire instrument was placed in a temperature-controlled, weatherproofed enclosure. This resulted in reproducible temperature profiles with minimal temperature gradients across the column (less than 2°C) during field operation (See Supplement Fig. S1).

### 2.2.4 Column Humidity Management

Because compounds are trapped at sub-ambient temperatures, relative humidity inside the column can easily reach 100% during ambient sampling. This is problematic not only because co-trapped water and ice clog the column, but also because many species of interest are highly reactive and can readily hydrolyze (Koppmann and Wildt, 2008; Roukos et al., 2009; Lee et al., 2014; Teng et al., 2017). We address this issue by diluting the ambient air with dry $N_2$ prior to cryotrapping to reduce

the RH below the ice point at -20°C (1.3hPa water vapor). This is illustrated in Fig. 3 during GC analysis of isoprene hydroxy nitrate (IHN) at high RH (~50%) with three different sample dilutions. When water is trapped during the lowest dilution (5x), the column flow is observed to decrease over time (Fig. 3A), indicating the formation of an ice blockage. In addition, the isomer distribution of IHN is dramatically altered, as seen by the loss of 1,2-IHN (first peak, Fig. 3D) and the corresponding formation of an isoprene diol, its hydrolysis product (Fig. 3G). However, at the two higher dilutions (15x and 20x), the column

flow remains relatively stable throughout the trapping period (Fig. 3B-C)—consistent with minimal ice formation—and the isomer distribution of IHN is preserved between the two runs (Fig. 3E-F). Some water is retained on the column even at these higher dilutions, but it was likely trapped downstream of the analytes, limiting its interactions with IHN.




During sampling, the operating dilution is chosen based on ambient relative humidity measurements. The effectiveness of the dilution is verified by monitoring the water signal (*m/z* 104) which should quickly fall to background levels during elution when minimal water is retained (as seen in Fig. 3E-F). For the data shown here, we diluted the samples by a factor of 15 during laboratory studies and by a factor of 20 to 30 in the field studies. The high sample dilution demands a very high sensitivity to

be able to adequately quantify many of the compounds of interest, which is achievable on this instrument due to the chemical ionization technique used (discussed below). Even so, ambient mixing ratios of several of the targeted analytes described here pushed the detection limits of the instrumentation, leading to increased uncertainty, especially when deconvolution is required prior to integration of chromatographic peaks.

## 2.3 GC/CIMS Interface

Following the column, a 100 - 200 $\mathrm{sccm}$ $N_2$ pickup flow (Fig. 1P) is added to the 5 $\mathrm{sccm}$ column flow to decrease the residence time in the PFA tubing connecting the GC to the mass spectrometer. A Teflon solenoid valve (225-T032, NResearch) then directs the analytes into the CIMS instrument, either through the flow tube (similar to direct CIMS sampling) or directly into the ion source. Unlike direct ambient sampling, it is possible to pass the GC flow through the ion source as oxygen is not retained on the column during trapping. Oxygen that enters the ion source is ionized ($O_2^-$) and causes interferences at many

*m/z*.

Figure 4 shows a comparison of two chromatograms obtained by these different analysis modes. Introduction via the flow tube (hereafter "FT" mode; Fig. 4, blue) allows for interaction of analytes with only $CF_3O^-$ (and $CF_3O^-$ derived) reagent ions, providing a straightforward comparison to the direct CIMS samples as well as quantification of the GC transmission of analytes. However, as the pressure within the column is greater under FT mode, due to tubing and gas flow configurations, than

when directed to the ion source region ($\Delta P$ =~30 $\mathrm{mbar}$), compounds tend to elute later and at higher temperatures, making introduction into the ion source (hereafter "high sensitivity" or "HS" mode; Fig. 4, black) the preferred analysis mode when separating more thermally-labile compounds.

HS mode also creates an enhancement in instrument sensitivity due to the increase in analyte-reagent ion interaction time. The enhancement in sensitivity is quantified through comparison to the direct CIMS measurements, which show a multiplica-

tive enhancement factor that is non-linearly dependent on the gas flow entering the ion source. For the instrument flows used in this work, the ion source enhancement was determined to be $9.8 \pm 0.8$ as calculated by methods described in the Supplement. Additional discrepancies between HS mode and direct CIMS measurements may result from analyte interactions with the metal walls of the ionizer. In addition, direct electron attachment to analytes (often followed by fragmentation) can also occur in the ion source. These fragment ions, however, provide additional structural information. For example, different fragment ions may

arise from the fragmentation of a primary nitrate versus a tertiary nitrate (see Supplement Fig. S5).

## 2.4 Instrument Housing and Supporting Equipment

The GC-HR-ToF-CIMS was placed in a weatherproofed, temperature-controlled enclosure during field sampling to protect the instrument electronics and allow for efficient GC cooling. In total, the instrument enclosure measured 1.1 m x 1.7 m x 0.9 m (W





x H x D), taking up a footprint of approximately 1 m$^2$ (Fig. 5). Weatherproofing was created by using Thermolite™ insulated paneling (Laminators, Inc.) that covered the aluminum instrument rack (80/20, Inc.) and was aided by weather stripping placed between the panels and the rack. For temperature control, two Ice Qube HVAC units (IQ1700B and IQ2700B, Blade series, cooling power = 498 and 791 W, respectively) were attached to one side of the enclosure to remove the heat produced by

the instrument. During the range of ambient temperatures experienced during these studies (8.7°C - 37.8°C), the internal temperature of the enclosure remained at or below 30°C under normal operating conditions.

Along with the instrument enclosure, two scroll pumps (nXDS 20i, Edwards) were located separately from the instrument in their own weather-resistant container and were used to back the three turbomolecular pumps (Twistorr 304 FS, Agilent) and the flow tube attached to the mass spectrometer. A weather station was also co-located with the instrument during the two field

studies. It included sensors for air temperature, RH, solar irradiance, wind direction, wind speed and atmospheric pressure.

## 2.5 Calibration and Instrumental Backgrounds

Calibrations were performed in the laboratory to measure the sensitivity of the instrument to a number of commercially available or synthesized standards. The absolute concentrations of these compounds were quantitatively determined by Fourier Transform Infrared Spectroscopy (FTIR) before being directed to the HR-ToF-CIMS (see Supplement for additional details regarding calibration procedures). However, as standards are not available for many species mentioned in this work, these

calibration experiments were simultaneously performed on the c-ToF-CIMS to directly compare the compound sensitivities between these two instruments. On average, the c-ToF-CIMS was 1.4 times more sensitive than the HR-ToF-CIMS for the species tested. We used this factor to proxy sensitivities for other compounds that were previously determined for the c-ToF-CIMS through calibrations or estimated using ion-molecule collision rates as described in Paulot et al. (2009a), Garden et al.

(2009), and Crounse et al. (2011).

We use two methods to quantify the instrumental background signals caused by interfering ions present at targeted analyte masses. In the first method, the instrument undergoes a "dry zero" where the CIMS flow tube is overfilled with dry nitrogen so that no ambient air is sampled during this time. In this method, the humidity within the instrument changes substantially compared with ambient measurements. The second method passes air from the main inlet through a zeroing assembly, which

includes a sodium bicarbonate denuder and a scrubber filled with Pd-coated alumina pellets. The scrubbed air then enters the flow tube after instrument flows are adjusted to mimic near-ambient humidity levels capturing an "ambient zero" which obtains background signals that are adjusted for the water dependent sensitivity of the compounds. During field sampling, both zeroing methods occur twice each hour during a six minute period that separates the CIMS and GC-CIMS measurements.

## 2.6 Data Processing

Data from the mass spectrometer is collected using data acquisition software provided by Tofwerk (TofDaq). This data is later combined with the instrument component read-backs collected using single board computers (Diamond Systems) and converted into a MATLAB file using in-house developed scripts. To account for fluctuations in the reagent ion, observed mass signals are normalized to the signal associated with the isotope of the reagent ion ($^{13}$CF$_3$O$^-$, *m/z* 86) and its cluster with water



$([H_2O \cdot {}^{13}CF_3O]^-$, *m/z* 104). The analyte signal is defined as this normalized absolute number of counts (nmcts) recorded at *m/z*.

### 2.6.1 GC Peak Integration & Identification

To integrate the chromatography peaks, we modified an open-source MATLAB peakfit function (O'Haver, 2017). Peak areas are determined for desired masses by subtracting a baseline and fitting the chromatograms with the appropriate peak shapes as shown in Fig. 6 for ISOPOOH and its isobaric oxidation product, isoprene epoxydiol (IEPOX, *m/z* 203; St Clair et al., 2016). For many compounds, preliminary peak assignment is based on previous laboratory studies that used a combination of chamber experiments and synthesized standards in order to determine elution order (Bates et al., 2014; Nguyen et al., 2014; Lee et al., 2014; Praske et al., 2015; Teng et al., 2015, 2017). However, due to differences in the analytical set ups, verification of these assignments and their retention times have also been made for a number of targeted compounds through laboratory experiments described in more detail in the Supplement. The results from one of these studies is shown in Fig. 7 which compares the retention times for alkyl hydroxy nitrates derived from propene (propene HN) and three structural isomers of butene (butene HN) created in the chamber bag with chromatograms gathered in the field.

## 3 Discussion

### 3.1 Analyte Transmission

The largest technical challenge in developing a field-deployable GC was the design of a sampling system capable of collecting and separating compounds with minimal analyte degradation. This is critical when considering that many targeted compounds are highly susceptible to irreversible losses or chemical conversion upon contact with instrument surfaces (Grossenbacher et al., 2001, 2004; Giacopelli et al., 2005; Rivera-Rios et al., 2014; Xiong et al., 2015; Mills et al., 2016; Hellén et al., 2017). We addressed this issue through the utilization of low pressure gas chromatography which holds several known advantages over traditional GC techniques (Sapozhnikova and Lehotay, 2015), such as creating conditions which allows compounds to elute both at lower temperatures and shorter retention times (Table 2). Lower elution temperatures better preserves thermally labile species and allows for the elution of lower volatility compounds within reasonable time scales. In addition, all wetted instrument surfaces (with the exception of the ion source) are composed of metal-free, inert materials such as PFA/PTFE Teflon, PEEK and column-phase materials. This reduces unwanted side reactions on surfaces, most notably the metal-catalyzed decomposition of compounds such as hydroxyperoxides and organic nitrates (Rivera-Rios et al., 2014; Mills et al., 2016).

Despite measures taken to improve analyte transmission, losses are still observed for some species, highlighting the importance of accurately quantifying analyte transmission through the GC column. Yet, for traditional GC-based measurements, transmission typically remains unknown which can be detrimental when there is a lack of available standards and GC response factors must be based on another compound that has a similar chemical make-up but may interact differently with the column phase. However, as previously stated, the combination of our LP-GC system with the high sensitivity of the CIMS provides





two sampling modes (direct CIMS and GC-CIMS) that automatically alternate between each other in half hour increments. This allows us to compare individual chromatograms to CIMS measurements taken immediately before or during cryotrapping in order to assess GC transmission efficiency under field conditions, without the need for external standards.

## 3.2 Sample Collection

Due to their lower volatility and highly reactive nature, the accuracy and precision of ambient OVOC measurements can be greatly limited by the sample collection method. GC sampling techniques typically used in atmospheric chemistry collect gas-phase compounds on solid adsorbents (e.g., TENAX®) that have been developed to combat some of the aforementioned issues (such as preventing the co-collection of water by allowing for higher trapping temperatures; Demeestere et al., 2007; Ras et al., 2009). However, the use of OVOC-specific adsorbents have shown problems with the formation of artifacts caused
by the reaction of ozone, $NO_2$, and other compounds trapped on the sorbent surfaces (Klenø et al., 2002; Noziére et al., 2015) and can lead to significant analyte loss, especially for polar and/or labile compounds such as tertiary organic nitrates (as suggested in Mills et al. (2016)), organic hydroperoxides and other highly-functionalized compounds. In addition, high humidities can result in increased water uptake into the sorbent materials (Ras et al., 2009) requiring additional water removal steps prior to collection such as trapping at above optimal temperatures which may result in the loss of more volatile compounds
(Vairavamurthy et al., 1992; Roukos et al., 2009) or through the utilization of chemical scrubbers which can react with intended compounds (Koppmann and Wildt, 2008; Roukos et al., 2009). These issues motivate our use of dilution and cryotrapping on the column to transmit a wider range of analytes through our system.

Trapping efficiency was assessed by cryofocusing a mixture of propene HN and IHN for varying amounts of time (and thus, sample volumes) in order to test for linearity of the cryotrap. Results provided in the Supplement show that the GC peak area
was linearly proportional to the volumes sampled suggesting that compounds are preserved on the column during trapping (Fig. S2). Analyte breakthrough has been monitored in the laboratory by directing the GC flow into the CIMS during trapping to monitor analyte signals. For most compounds of interest (>$C_3$), there has been no evidence of breakthrough under typical trapping conditions (-20°C) when this procedure has been performed for a trapping period up to 12 minutes.

Our trapping temperature (-20°C) was optimized on the original test bed and was chosen as the best compromise for its
ability to capture compounds with a range of volatilities at the highest possible temperature and, thereby, the lowest dilution required to avoid trapping water. We find that trapping above -20°C results in degradation of the chromatography for several species, examples of which can be seen in the Supplement (Fig. S3). However, even at -20°C some higher volatility compounds are still not trapped efficiently, resulting in irregular peak shapes (Fig. S4). Further optimization of trapping conditions is needed in order to improve the chromatography for these species and further reduce the likelihood of water co-trapping.

## 30 4 Field Performance and Ambient Air Measurements

The GC-HR-ToF-CIMS has participated in two field studies that served as a test for this analytical method. Its first deployment occurred as part of the Program for Research on Oxidants, Photochemistry, Emissions and Transport (PROPHET) campaign



in summer 2016, where it was placed on the top of a 30 m research tower surrounded by the dense forests of rural, northern Michigan. The following summer, the instrument underwent a second deployment at the California Institute of Technology (Caltech) campus in Pasadena, CA and sampled from the roof of the 44 m tall Millikan Library. In contrast to PROPHET, Pasadena is typically characterized as a high-$NO_x$, urban environment due to its proximity to Los Angeles. However, biogenic

emissions have also been known to influence the area (Arey et al., 1995; Pollack et al., 2013), due to local urban flora and the presence of the San Gabriel Mountains to the north.

During both deployments, the instrument provided a near continuous measure of OVOC concentrations, though we experienced occasional interruptions in the GC measurements at both locations due to required maintenance of the cooling system. However, instrument upgrades performed prior to the Caltech study were able to greatly reduce GC downtime and significantly

improved the chromatography, despite other operating conditions remaining mostly unchanged. When the GC was operational, data was captured during 1 h cycles in which the first half was dedicated to direct CIMS measurements and the latter half measured analytes after chromatographic separation, with the collection of ambient and dry zeros interlaced between operational modes. This sampling routine is shown in Fig. 8 for a single mass (*m/z* 232) collected during the 2017 Caltech field study.

The data sets described here focus on the daytime isoprene degradation products such as IHN, ISOPOOH, IEPOX and

HPALD. These species are chosen because they are unique to the isoprene oxidation pathways, allowing for a more complete analysis for the atmospheric production and fate of each isomer. At PROPHET, products from the $HO_2$ reaction pathway (ISOPOOH and IEPOX) were the most abundant among the discussed species, reaching an average maximum of ~200 pptv during a three day period (Fig. 9A). Because ISOPOOH and IEPOX are mass analogues, most analytical techniques are either unable to separate these two species or rely on the relative abundances of fragment ions to determine the relative contribution

of each to the observed signal (Paulot et al., 2009b). With the GC-CIMS, however, we are able to physically separate the isomers prior to quantification (Fig. 6). As seen in Fig. 9, IEPOX comprised about half of the total daytime signal (07:00 - 22:00 local time) with an average *trans:cis* ratio of 1.7. For the ISOPOOH isomers, an average daytime 1,2-ISOPOOH to 4,3-ISOPOOH ratio of ~7.6 was observed. The ISOPOOH isomer ratio is much higher than expected accounting only for the isomer-specific bimolecular reaction rates of the isoprene peroxy radicals (Wennberg et al., 2018). The higher ratio is consistent

with a large sink of the 4-OH $RO_2$ isomers via $RO_2$ isomerization (Peeters et al., 2009; Crounse et al., 2011; Teng et al., 2017). The importance of such unimolecular chemistry is further supported by observations of known isomerization products (e.g. HPALDs; Fig. 10) found throughout the course of the campaign.

IHN was also observed at PROPHET, though in much lesser amounts than ISOPOOH or IEPOX. Only two isomers could be identified in the GC data collected during this experiment: 1,2-IHN and 4,3-IHN with an average daytime ratio of ~2.6.

We compare these IHN observations from PROPHET to measurements from the Caltech site to assess differences in $RO_2$ chemistry between the two sites. Similar to PROPHET, 1,2-IHN and 4,3-IHN were the first and second most abundant isomers of this compound observed at the Caltech site, respectively, though in this study other IHN isomers were also quantified (Fig. 11), as well as an unidentified component that has been previously observed during laboratory studies (Teng et al., 2017). During the Caltech study, the average daytime 1,2-IHN to 4,3-IHN ratio was ~1.4, roughly half that observed in Michigan;

we suspect this difference reflects the shorter bimolecular lifetime of the ISOPOO in Pasadena (<10 s) which would limit



the impact of the competitive unimolecular reaction pathways in this environment. Interestingly, the IHN ratio at PROPHET differed significantly from the corresponding ISOPOOH ratio despite the similar formation pathways of each pair of oxidation products. We suspect this reflects differences in their loss pathways and hypothesize that the lower isomer ratio for the pair of nitrates may result from hydrolysis of the 1,2-IHN isomer (see also Wolfe et al., 2015; Fisher et al., 2016).

In addition to daytime isoprene oxidation products, the GC-CIMS captured a wide variety of additional compounds, some of which can be identified based on previous laboratory studies. For example, evidence of isoprene + $NO_3$ chemistry at Caltech is indicated by the increase in the signal at *m/z* 230, which is assigned to the isoprene carbonyl nitrates (ICN). Though only two isomers were observed during this study (Fig. 12), the distribution of these species (assigned as 4,1-ICN and 1,4-ICN) matches results from Schwantes et al. (2015) and is consistent with their finding that C1 addition of the $NO_3$ moiety is favored

(Suh et al., 2001). As the distribution of the isoprene nitroxy peroxy radical ($INO_2$) is less constrained than the OH derived $RO_2$ counterpart, further observations of ambient ICN isomers with the GC-CIMS may lead to improved understanding of the impact of nighttime $NO_3$ chemistry and provide additional information on the relative importance of ICN degradation pathways (e.g. photooxidation) and thus its effect on $NO_x$ concentrations at sunrise (Müller et al., 2014; Schwantes et al., 2015).

Through the combination of chromatographic separation, high mass resolution and low-fragmentation mass spectrometry the GC-HR-ToF-CIMS will serve as a powerful tool, helping to untangle the atmospheric chemistry of many OVOCs. This is illustrated in observations of several presently unidentified compounds measured during the field studies, such as *m/z* 236 (MW 151), a suspected nitrogen-containing compound observed at Caltech (Fig. 13). Data obtained from direct CIMS sampling showed at least two local maxima, one occurring before sunrise and the other shortly after noon. With the addition of the GC,

we find that two distinct species contribute to this instrument signal with varying contributions over the course of a day. That is, the first compound (eluting at 9.8 minutes) is responsible for the majority of the signal in the early afternoon, possibly indicative of production via photooxidation, whereas the second compound (eluting at 13.8 minutes) is most abundant between sunset and sunrise, possibly due to production from nighttime $NO_3$ chemistry, high photolability, a short lifetime against the OH radical, or some combination thereof.

## 25   5   Summary

We have developed an automated GC-CIMS system that can capture diurnal changes in the isomer distributions of a wide range of important OVOCs. This novel method addresses common issues typically associated with ambient GC measurements, allowing observations of compounds that have previously proven difficult to measure. We use a combination of sample dilution and temperature control to avoid the adverse effects caused by high column humidity (e.g. hydrolysis of reactive compounds).

This, along with the use of LP-GC methodology, cryotrapping directly on the column and the creation of a mostly metal-free GC design, reduces analyte degradation upon contact with the instrument surfaces.

Analytical performance was assessed through a combination of laboratory studies and field campaigns. GC-HR-ToF-CIMS has demonstrated its ability to provide reproducible measurements, effectively trapping tested species with no observable



breakthrough and providing a quantitative measurement of GC transmission. Though additional optimization is needed to expand the number of species that can be measured using this technique, its participation in future field studies will help enable the elucidation of the chemical mechanisms of a number of species, such as the isoprene oxidation products, by providing information that will help assess how compound structure impacts its atmospheric fate and thereby its effect on the global

5   atmosphere.

*Data availability.* Data from the 2017 Caltech study is available at http://dx.doi.org/10.22002/D1.971. Additional data is available upon request to the corresponding authors.

*Competing interests.* The authors declare they have no conflict of interest

*Acknowledgements.* We would like to thank Steve Bertman, Phil Stevens, and the University of Michigan Biological Station (UMBS) for

10   organizing the PROPHET 2016 campaign. We are indebted to the many PROPHET participants who helped us move the instrument to and from the top of the tower safely. We also thank the Caltech campus and affiliated staff for accommodating the Summer 2017 study. The construction of the GC-HR-ToF-CIMS was supported by the National Science Foundation (AGS-1428482), with additional NSF support (AGS-1240604) provided for the instrument field deployments. Work performed by KTV and HMA was supported by the National Science Foundation Graduate Research Fellowship (NSF GRFP). KTV also acknowledges support from an Earl C. Anthony Fellowship in Chemistry

15   during an early portion of this study.



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



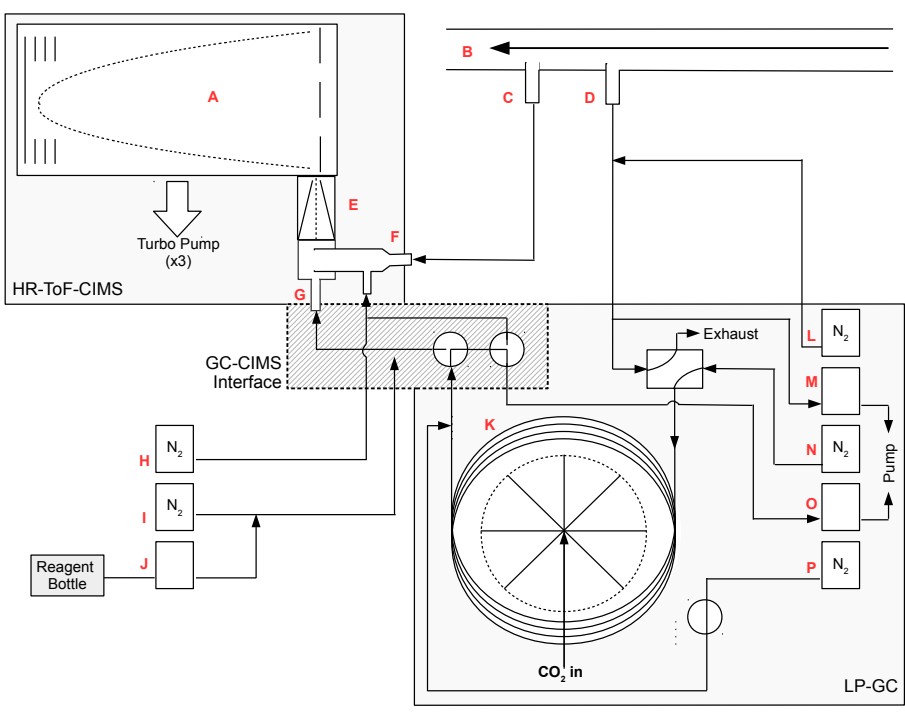

**Figure 1.** A simplified instrument schematic of GC-HR-ToF-CIMS showing the HR-ToF-CIMS, the LP-GC and the interface between the two systems. Main components are: (A) time-of-flight mass spectrometer; (B) teflon coated glass inlet; (C) CIMS sampling port; (D) GC-CIMS sampling port; (E) hexapole ion guide; (F) teflon coated glass flow tube; (G) 210-Po ionization source; (H) CIMS dilution flow; (I) CIMS ion source dilution flow; (J) $CF_3OOCF_3$ reagent flow; (K) GC column and cryotrap; (L) GC dilution flow; (M) GC sample intake pump; (N) GC column flow; (O) GC bypass pump; (P) GC $N_2$ pickup flow. Diagram is not to scale.




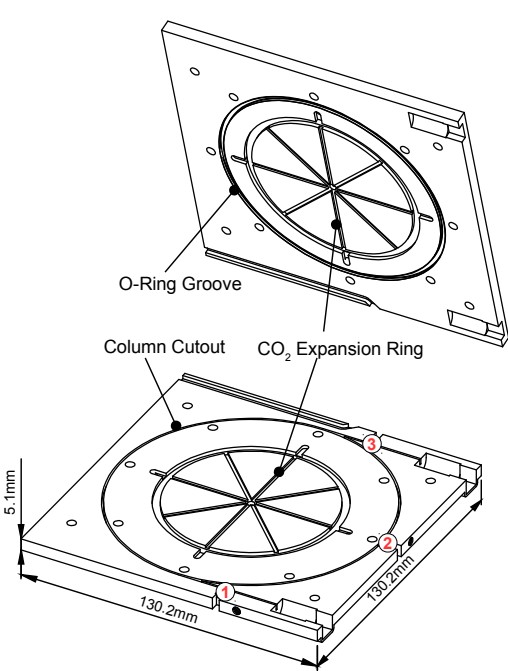

**Figure 2.** Schematic of the GC cyrotrap and heating unit. Column sits in a groove machined into one plate, providing good thermal contact. $CO_2$ enters from the center of both plates (on the opposite side) and expands in the eight radial spokes before exiting through four exhaust ports (opposite side). The temperature is measured at three locations near the column: (1) near the inlet of the column, (2) on the column ring, and (3) near the outlet of the column.



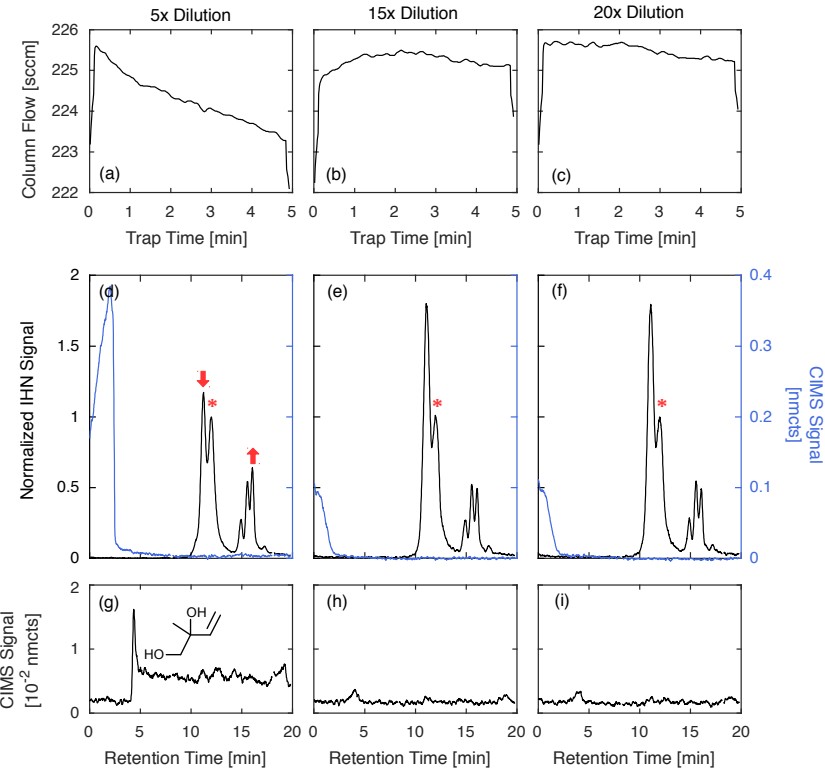

**Figure 3.** Comparison of GC column flow (A-C) and three chromatograms (D-F) of IHN (*m/z* 232, black) and water (*m/z* 104, blue) at three different dilutions from a high RH chamber experiment. The beginning of a chromatogram is marked when the temperature program initiates. When water is trapped during the lowest dilution (5x), column flow decreases (indicating an ice blockage) and the isomer distribution of IHN is dramatically altered as noted by a loss in the first peak (1,2-IHN) and increase in the last peak (*E* 1,4-IHN). These peak changes are marked by arrows and described relative to 4,3-IHN (*). The 1,2-isoprene diol (*m/z* 187, G), an expected product of 1,2-IHN hydrolysis, is also observed in this scenario. However, when the sample is sufficiently diluted prior to trapping, the water signal quickly falls to background levels and isomer distribution is preserved with minimal diol formation. Column flow also remains relatively stable throughout the trapping period when minimal water is retained.





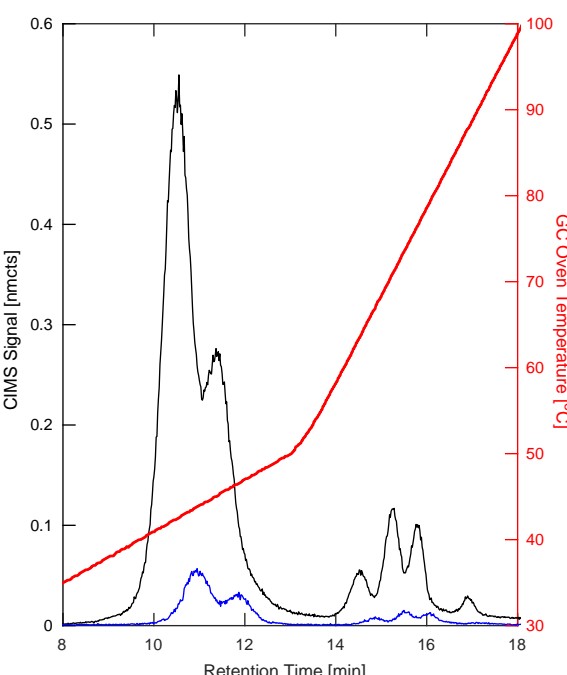

**Figure 4.** Comparison of chromatograms of the IHN isomers obtained from the two different GC analysis modes in which the same amount of analyte is collected on the column, but is directed into either the ion source (black) or flow tube (blue). GCs that are directed into the ion source result in approximately a 10-fold signal increase compared to flow tube GC analysis. In addition, compounds analyzed via the ion source typically elute at lower temperatures compared to flow tube analysis, an advantage for sampling fragile, multifunctional compounds.



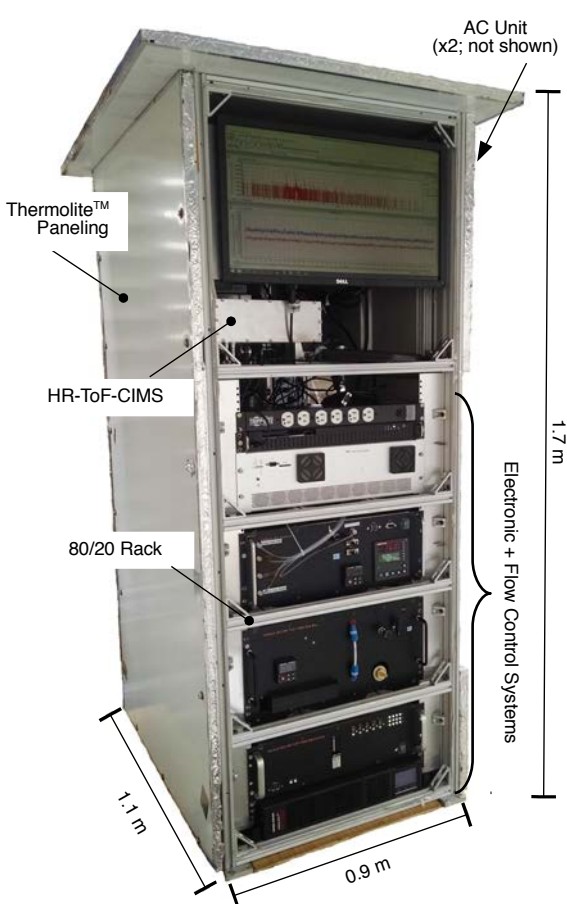

**Figure 5.** The weatherproofed and temperature-controlled enclosure in which the instrument resides during field sampling. The front panel of the enclosure is removed in this photo.



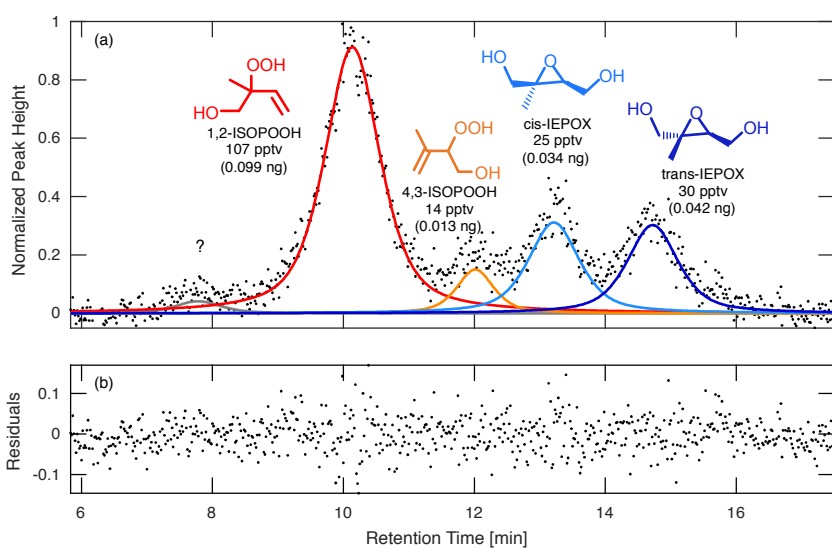

**Figure 6.** (A) Chromatogram, peak fits and (B) fit residuals resulting obtained from the peakfit MATLAB function for the deconvolution and integration of ambient ISOPOOH and IEPOX isomers observed during the PROPHET 2016 field study. The isomers observed during this study were 1,2-ISOPOOH (red), 4,3-ISOPOOH (orange), cis-IEPOX (light blue) and trans-IEPOX (dark blue). In addition, an unknown peak (gray) can be seen eluting at 7.8 minutes prior to the ISOPOOH and IEPOX isomer species. To obtain the ambient mixing ratios, peaks are deconvoluted and integrated using an appropriate peak shape (in this case, a Gaussian-Lorentzian blend), scaled by the relative CIMS sensitives of each isomer (see Supplement), ion source enhancement (if applicable) and estimated transmission factor, and then normalized by volume of air collected on the column. The GC signal shown here has been normalized to the largest peak height. Amounts shown in parenthesis corresponds to the amount of analyte trapped in the column.





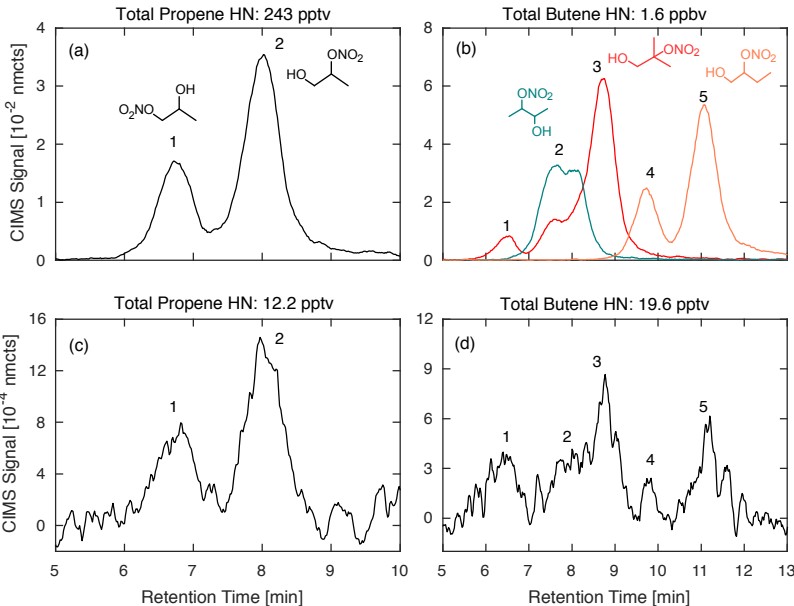

**Figure 7.** Comparison of hydroxy nitrates formed during chamber experiments (A-B) from propene (left) and three structural isomers of butene (right; 1-butene (orange), 2-butene (teal), and 2-methyl-propene (red); dominant hydroxynitrate structures shown) with the corresponding *m/z* signal observed during a 2017 field study in Pasadena, CA (C-D). Data shown is a 10 second average.



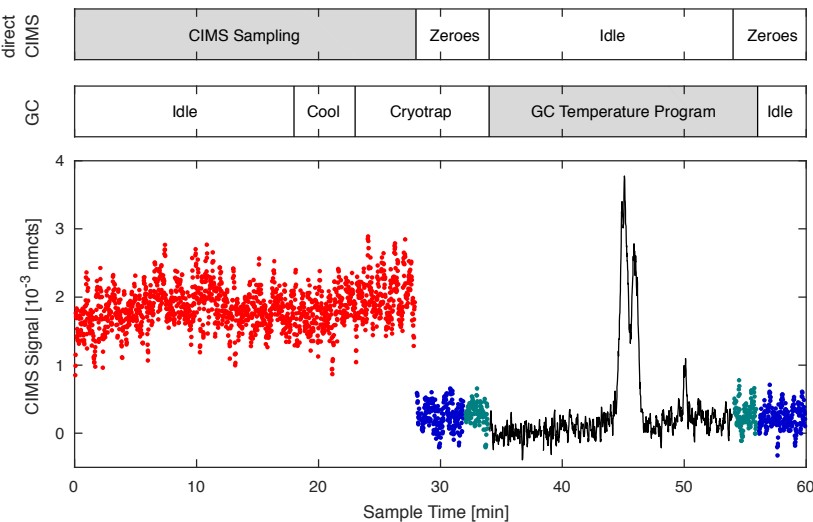

**Figure 8.** Typical GC-CIMS sampling cycle during the 2017 field study in Pasadena, CA. Data shown for *m/z* 232. Cycle has a period of 1 hour in which the first half is dedicated to direct CIMS measurements (red), the latter half measures compound signals that have undergone chromatographic separation (black). The two sampling modes are separated by a zeroing periods comprised of a four minute ambient zero (blue) and a two minute dry zero (green). Most GC processes occur in the background during direct sampling, as to not interrupt data collection. Data shown here is a two second average. Changes in the amount of flow entering the ion source during direct CIMS and GC-CIMS sampling directly correlate with the signal to noise seen during each operating mode. The increased flow rate through the ion source during the GC sampling mode results in higher ion counts and increased signal to noise.





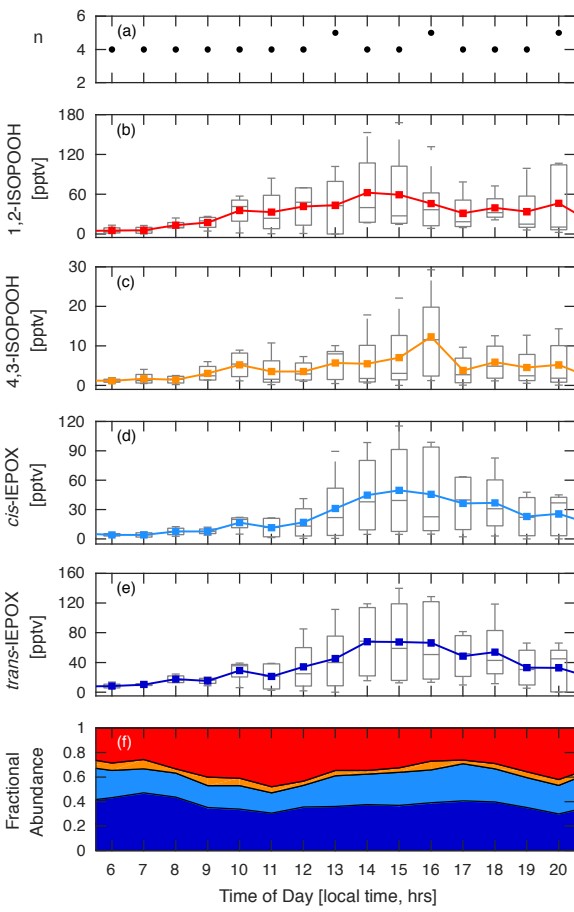

**Figure 9.** Average (mean) diurnal profiles obtained from *n* number of chromatograms (A) collected during the PROPHET campaign for (B) 1,2-ISOPOOH, (C) 4,3-ISOPOOH, (D) *cis*-IEPOX, and (E) *trans*-IEPOX (marked by colored squares). Data was collected between 23 - 28 July, 2016. For each box surrounding these average values, the central lines mark the median, the top and bottom edges represent the 25th and 75th percentiles, respectively, and the whiskers mark the maximum and minimum values observed that are not considered outliers (marked separately by a red '+' symbol). (F) Average diurnal profile of the fractional abundance of each of these four isomers based on their mean values.





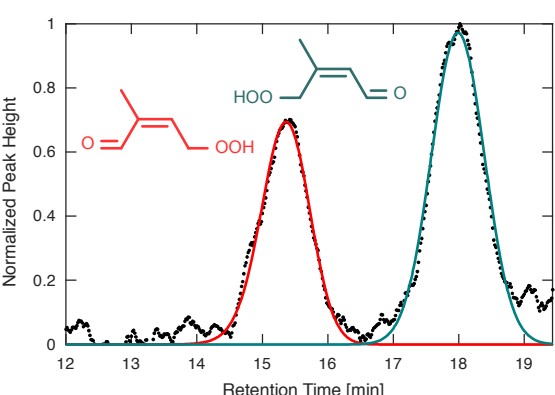

**Figure 10.** Chromatogram obtained during the PROPHET campaign for the two HPALD isomers providing evidence of $RO_2$ isomerization in that environment. GC signal has been normalized to the largest peak height.




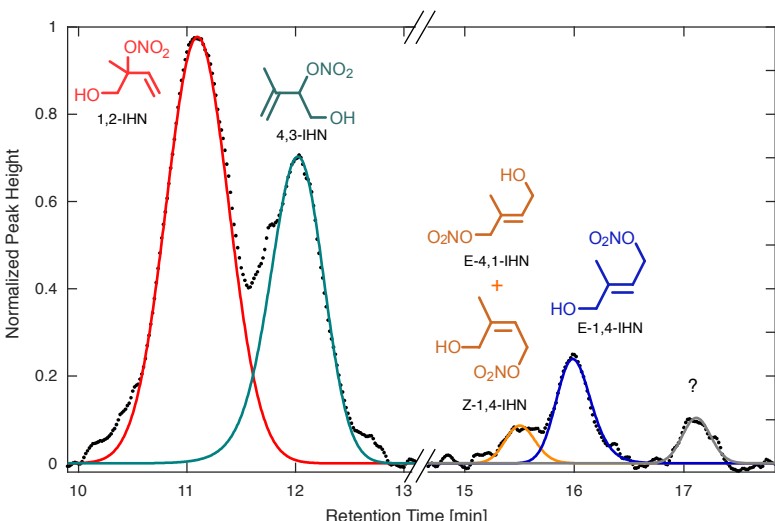

**Figure 11.** Chromatogram obtained during the Caltech field study for *m/z* 232, attributed to the IHN isomers, normalized to largest peak height. At least four isomers of IHN were observed: 1,2-IHN (red), 4,3-IHN (green), *E*-4,1- and *Z*-1,4-IHN (coelute, orange), and *E*-1,4-IHN (blue). *Z*-4,1-IHN was not present above the instrument detection limit. An unidentified component, which likely corresponds to a species observed in laboratory isoprene oxidation studies, is present near the end of the chromatogram (grey, see Teng et al. (2017)).



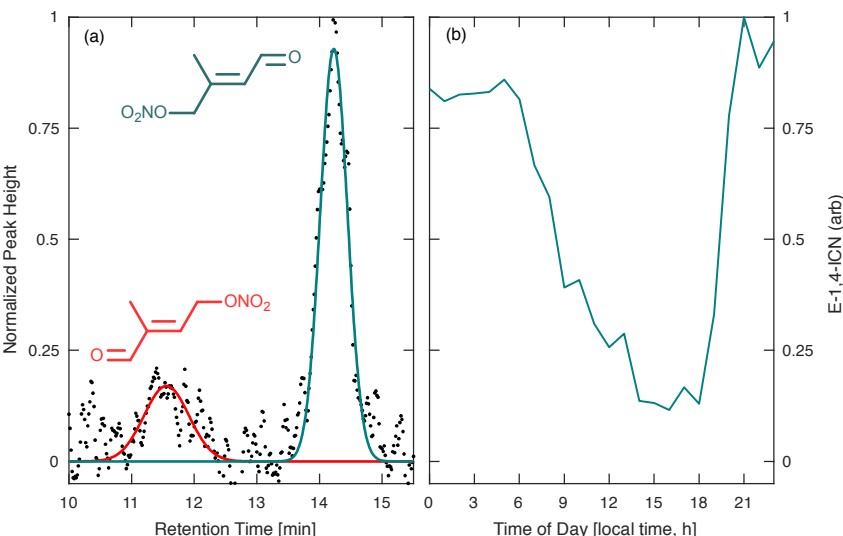

**Figure 12.** (A) Chromatogram obtained during the Caltech field study for the two isoprene carbonyl nitrate isomers (4,1-ICN in red and 1,4-ICN in green, *m/z* 230) produced by isoprene + $NO_3$ chemistry, normalized to the largest peak height. Peak assignment is based on results from Schwantes et al. (2015). (B) Average diurnal profile of most abundant ICN isomer, 1,4-ICN, obtained from chromatograms collected between 01-16 Aug, 2017 during the Caltech field study. This profile appears to correspond with the expected formation of ICN from $NO_3$ oxidation of isoprene in dark/dim conditions and the rapid loss in light periods.





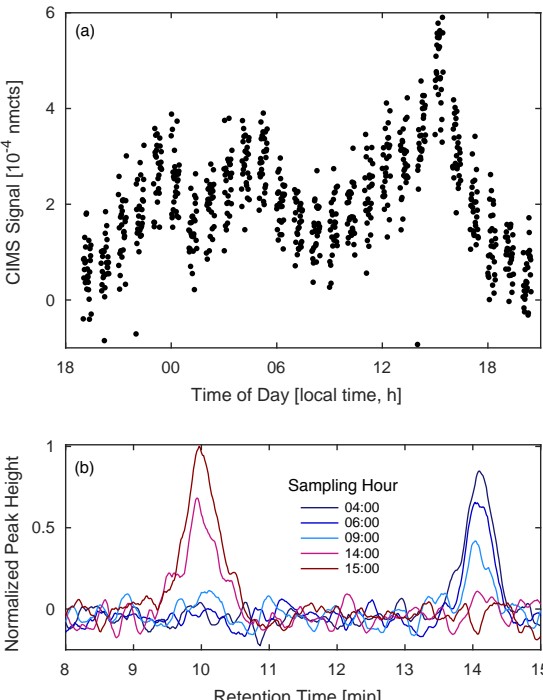

**Figure 13.** (A) Diurnal profile of unidentified compounds observed at *m/z* 236 (MW 151) from 11-12 Aug, 2017 during the Caltech field study and (B) select field chromatograms from the same sampling period. The GC shows at least two compounds contribute to the signal, one more abundant at night (blue) and the other more abundant in the late afternoon (red).



**Table 1.** Examples of OVOCs measured in this study.

| Compound | Abbreviation | Example Structure |
|---|---|---|
| isoprene hydroxy nitrate | IHN | 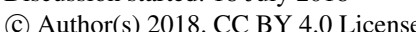 |
| isoprene hydroxy hydroperoxide | ISOPOOH | |
| isoprene epoxydiol | IEPOX | (cis) |
| isoprene hydroperoxy aldehyde | HPALD | |
| isoprene carbonyl nitrate | ICN | |
| propene hydroxy nitrate | Propene HN | |
| butene hydroxy nitrate | Butene HN | |
| propanone nitrate | PROPNN | |
| hydroxymethyl hydroperoxide | HMHP | |



**Table 2.** Comparison of elution temperature (°C) and retention time (minutes, in parenthesis) for isoprene nitrates.

| Study | Column | 1-OH 2-N | 4-OH 3-N | $Z$ 4-OH 1-N | $E$ 4-OH 1-N | $Z$ 1-OH 4-N | $E$ 1-OH 4-N |
|---|---|---|---|---|---|---|---|
| Mills et al. (2016) | Rtx-1701[a] | N/A | 110 (26.1) | 119.2 (36.5) | 133.7 (39.3) | 133.2 (39.4) | 142.7 (41.2) |
| Mills et al. (2016) | Rtx-200[a] | N/A | 101.1 (16.7) | 110 (22.4) | 110 (25.1) | 110 (23.3) | 110 (26.5) |
| This Study | Rtx-1701[b] | 42.4 (10.5) | 45.1 (11.4) | 63.2 (14.5) | 71.3 (15.3) | 71.3 (15.3) | 76.4 (15.8) |

[a] Column is 30 m, 0.32 mm ID, 1 μm phase thickness

[b] Column is 1 m, 0.53 mm ID, 3 μm phase thickeness