# Peer review of "Low-pressure gas chromatography with chemical ionization mass spectrometry for quantification of multifunctional organic compounds in the atmosphere"

_Atmospheric Measurement Techniques, 2018_

## Referee Comment (RC1) · Anonymous Referee #1 · 7 Sep 2018

Reviewer comments on Vaquez et al., "Low-pressure gas chromatography with chemical ionization mass spectrometry for quantification of multifunctional organic compounds in the atmosphere"

The authors present here a new instrument for the chromatographic separation and analysis of labile compounds of interest to the atmospheric chemistry community, in particular many known oxidation products of isoprene. The instrument is technically sound, relatively thoroughly described, and has the potential to significantly advance our understanding of the atmospherically-important isoprene oxidation system. I have

only minor concerns regarding the scientific or technical aspects of this work, but find there are a few larger issues regarding the overall presentation, leading me to recommend publication following major revisions addressing the comments below.

General comments:

1) Sections of the instrument description are not clear. For example, from the diagram in Figure 1 it is not clear to me what the sample flow path is or where the sample is actually collected. Is it pulled straight through the entire column, or is it trapped on the 4-port valve? What is the purpose of pump M? What is the purpose of flow tube F? A few more of these sorts of questions come up throughout the comments below.

2) Sections of the instrument calibration are not clear. For example, it is mentioned that some compounds have sampling losses, but then there is no further mention or discussion of this. Calibration appears to rely in some way on a different c-ToF, but exactly how that is being used is not clear. It sounds like not all compounds have authentic/synthesized standards, and if not, how are retention times determined? Estimating collection and transfer efficiency by comparing GC-CIMS signal to CIMS signal likely suffers from an assumption of equal transmission of all isomers, but it is not clear if that is being accounted for because there is little discussion of how it is being applied.

3) Demonstrating the capabilities by field deployment is a valuable addition, but the authors seem to be focusing more on the actual isoprene oxidation chemistry and science than in thinking about this as a proof-of-concept. Several detailed chemical questions regarding unimolecular reactions, etc., are addressed and discussed all within a few short paragraphs, which does relatively little to advance the instrumentation aspect of the paper, but is also too detailed and dense to give a good discussion of the scientific advances. My strong recommendation is focus on this section only as a proof-of-concept. For instance, a major advantage of this technique is the time-resolved datayet the authors show no timeseries of concentrations or isomer ratios, or demonstrations of the continuous operation other than inferring it from the diurnal patterns.

Overall, all three comments are sort of different sides of the same coin - my impression is that the authors seem to be skipping past or over some key instrument issues (in part by relying heavily on the supplementary information) in order to get to the science, and in doing so are not quite giving either side of the story the attention they deserve. I think the approach is sound, believe it is well tested, and am excited by the scientific prospects, but I think better organization and focus would better serve both the instrument descriptions and the science.

Technical comments:

Page 2 line 3: "can also lead to a scenario" sounds a bit odd to me, maybe just "Chemical oxidation can cause OVOCs to increase....".

Page 2 line 5: Use "In addition" or "also", but probably not both

Page 2 line 27: The GC paragraph sounds like it is about GC in general, in which case it would be a much larger discussion. I think the authors are mostly discussing field-deployabe/in-situ GCs here, which should be made clear.

Page 2 line 30-31: Though I realize it is not possible to make this list of GC-based OVOC measurements comprehensive, there are a few absences that stand out. I would include some of Allen Goldstein's measurements in this list, perhaps Millet et al., JGR 2005 which saw MVK and methacrolein as well as other OVOCs, and arguably SV-TAG as I believe it can see many oxygenated gases (Zhao et al., AS&T 2013). I would also include the NOAA GC, such as Goldan et al., JGR 2004, and/or some of Jessica Gilman's work.

Page 3 line 22: I don't understand how the air is subsampled. Is there a sample loop or something? Oh, on the head of the column - so is there a valve in the CIMS interface? There needs to be some more clarity on how sampling happens

Page 3 line 17: What is "Low pressure" about it? It seems like a regular GC approach to me: trapping cold, then heating and pushing through N2 to a vacuum detector. I

gather the "LP" aspect is the large bore column, which reduces carrier gas pressures? A short mention or discussion of this would be helpful.

Page 3 line 28: The compounds aren't separated by the temperature controller, they are separated by the GC, using a ramp controlled by the controller.

Page 3 line 29: "each plate" is not that clear. Do the author's mean "on either side of the GC manifold/housing"?

Page 3 line 30: What is the purpose of the flow tube? It makes a big difference later, but it's not clear what the function is. I would think interaction with ions, but seems to happen latter.

Page 6 line 16-21: FT and HS is assymmetric naming, with one after the approach and the other after the result. FT and IS (ion source) would be preferred,

Page 6 line 23: It's not clear why introduction at the source causes longer interaction times. Does fragmentation affect/complicate calibration?

Page 7 line 14: When the authors say "directed" do they mean direct sampling, or analysis by GC? Given the fragmentation in the HS method, it seems to me that the latter is necessary.

Page 7 line 15: Which standards are available/synthesized and which are not? Later, for instance in Figures 7, 10, and 11, the author's seem to know the elution orders of many specific isomers - are these all from authentic or synthesized standards?

Page 7 line 16-20: It is not clear what the purpose of the c-ToF-CIMS is, or where it is discussed or first mentioned. Is it that there are known sensitivities with that instrument, so that is the purpose of the average sensitivity difference? How does it help to compare to the c-ToF? This paragraph needs generally more explanation to be made clearer.

Page 8 line 27: for which species are losses observed?

[Figure]

Page 9 line 2-3: This is an interesting approach that potentially provides very nice con-
firmation of compounds for which standards aren't available. However, do the author's
know that all isomers are transferred equivalently, and that all isomers can be seen?
If total isomer-resolved signal is less than direct CIMS signal, that could be due to
incomplete transmission of all isomers as seems to be implied by the author's, but it
could also be due to complete transmission of one isomer but not the others, or com-
plete transmission of all observed isomers but the presence of non- or poorly-observed
other isomers. It is not clear to me that this approach fully works, and no effort is made
to validate it here.

Page 9 Section 3.2: Dilution will solve the humidity problem, but only reduces and does
not solve the problem of reactions on the adsorbent from ozone or other oxidants. To
collect these compounds, an ozone scrubber is probably out of the question, but have
the author's done any tests to evaluate the impact of ozone on these compounds under
typical sampling conditions?

Page 10, Section 4: How long were these campaigns and/or measurment periods?

Page 10, line 18: "Isomers" is a more common term than "mass analogous", or the
authors could use "isobaric", the mass spectrometric term for having the same mass

Figure 3d-f: Both axes are CIMS signals, but one is labeled normalized IHN signal, and
the other as CIMS signal. Why not label the right as m/z 104 signal, or water signal or
something comparable?

Figure 8: The period of GC elution (black line) seems to have significantly lower scatter,
even during background periods - is that real, and if so why is that?

———————————————————

---

## Referee Comment (RC2) · Anonymous Referee #2 · 3 Oct 2018

**Low-pressure gas chromatography with chemical ionization mass spectrometry for quantification of multifunctional organic compounds in the atmosphere – Vasquez et al.**

This paper will be an important contribution to the literature. It is mostly well-written although occasionally too indirect and not descriptive enough in describing key features of the instrument. The introduction is excellent – spot on. An excellent discussion is given in Section 4 regarding the field deployment of the instrument and its utility in ambient air measurements in forested environments. The instrument is very impressive and the authors did some fantastic work in developing this technique.

However, after the introduction, the paper loses focus and does not accomplish the objective of properly describing an analytical instrument. It is stated that the novelty of the paper comes from the "field-hardened design" implying that the paper is focused on that. The reader is presumably supposed to look back at previous publications from the group to fill in the gaps and missing details from this paper and be satisfied that the description of the field – hardened instrument is sufficient. This reviewer is too lazy to scour through those previous publications and would like this paper to stand-alone. I think that's a reasonable expectation that could be met fairly easily by the authors

**Instrument description section:**

Suggest devoting significantly more time on the first paragraph describing the instrument and Figure 1. Even if the details of this are in previous papers. The title speaks of low pressure chromatography but the words "low pressure" are mentioned only two times and with very little description of it, how it works, what pressures the GC operates under etc.

It is not clear at all how the cryofocusing is accomplished. Please clarify this section and take the time and space to describe the different parts of Figure 1 – particularly the cryofocus and low pressure aspects of the GC. The very high flow rates are an interesting aspect of the design and this should be highlighted and explained.

Since this paper is about the description of an automated field-hardened instrument, provide more details on how the various components of the instrument are fitted together and how the automation was accomplished. The description of the instrument is not concise and does not have a good flow.

Often very indirect language is used which results in the manuscript being too wordy perhaps at the expense of not providing concise details. An example:

5/11: "During the collection of analytes on the head of the column, it is important that the temperature remains stable, as sizable fluctuations in temperature adversely affects the chromatography. To control the trapping set point…"

*Could be replaced by something like:*
A PID control loop using heaters and the resistance temperature detector (RTD, F3102, Omega) located on the GC column ring (Fig. 2, #2 on diagram) were used to maintain fine control over the temperature set points during cryofocusing. This is needed to obtain reproducible chromatography.

Suggest going through the Instrument description sections and make clear declarative statements where possible and appropriate of the instrument design. Provide details needed for the reader to grasp the primary design features and justification for them without having to refer to previous papers.

**Calibrations and backgrounds**

7/15: However, as standards are not available for many species mentioned in this work, these calibration experiments were simultaneously performed on the c-ToF-CIMS to directly compare the compound sensitivities between these two instruments. On average, the c-ToF-CIMS was 1.4 times more sensitive…

I know what you mean here and it is explained further in the supplement but please rewrite more clearly in the main section as other readers will not get this on a quick read through.

7/21:

We use two methods to quantify the instrumental background signals caused by interfering ions present at targeted analyte masses. In the first method, the instrument undergoes a "dry zero" where the CIMS flow tube is overfilled with dry nitrogen so that no ambient air is sampled during this time. In this method, the humidity within the instrument changes substantially compared with ambient measurements. The second method passes….

How do the two methods compare?

5/12: To control the trapping set point, we utilize the heaters and the resistance temperature detector (RTD, F3102, Omega) located on the GC column ring (Fig. 2, #2 on diagram)

Perhaps show the heaters on the diagram

5/14: In addition, during trapping we only use the solenoid valve connected to the 0.15 mm I.D. restrictor as this valve provides a $CO_2$ flow that is adequate to maintain the GC temperature (~10 slm) ?????

**Discussion**

8/16: The largest technical challenge in developing a field-deployable GC was the design of a sampling system capable of collecting and separating compounds with minimal analyte degradation.

Why is this true for a field-deployable system? Seems that you need those same characteristics for a laboratory-based system. The difference in a field – deployable system one would think is in getting the sample undisturbed to the instrument which is not addressed – and possibly trivial if the right sampling manifold is used (also not discussed). Referring to my opening comments, the question here is whether more of the details of the system – or a similar prototype system are discussed in previous papers. I suggest that these details be repeated here for the reader. Address what was specifically done in the field-deployable GC versus the prototype laboratory system.

**Field Performance:**

10/9: "However, instrument upgrades performed prior to the Caltech study were able to greatly reduce GC downtime and significantly improved the chromatography, despite other operating conditions remaining mostly unchanged."

This in a nutshell exemplifies the main problem with the paper. What were the instrument upgrades? Isn't this what the paper is supposed to be about?

Figures:

Fig 1. Enlarge the LP-GC portion of the drawing with better detail on the valving and cryofocusing aspects
Fig 2. Enlarge drawing and add heaters on solenoid positions
Figures 3-13 are good and appropriate

Small thing…

4/22: For the studies detailed in this paper…unnecessary to start the sentence with this. Check paper for other such incidences

---

## Author Comment (AC1) · 31 Oct 2018

Thank you for the helpful comments. Our responses can be found below.

General Comments:

1) Sections of the instrument description are not clear. For example, from the diagram in Figure 1 it is not clear to me what the sample flow path is or where the sample is actually collected. Is it pulled straight through the entire column, or is it trapped on the 4-port valve? What is the purpose of pump M? What is the purpose of flow tube F? A

few more of these sorts of questions come up throughout the comments below"

We have made the following changes in order to improve the instrument description section:

* Fig. 1 has been updated so that the instrument flow paths are differentiated from each other. We now highlight the flow paths of direct CIMS sampling, GC trapping, and GC elution. We believe this makes it clear that diluted air is pulled through the column during trapping. This update also makes it clear that the GC pumps either pull air from the instrument inlet or pull air through the column for sample collection.

* We have included an improved description on how air is subsampled into the column while referencing specific components labeled in Fig. 1 in order to better describe their purpose. An example of such changes to the manuscript can be seen on page 5 lines 19-22.

* The purpose of the flow tube is to ensure the analyte stream is well mixed prior to chemical ionization by CF3O-. This has been clarified on page 4 lines 3-5.

2) Sections of the instrument calibration are not clear. For example, it is mentioned that some compounds have sampling losses, but then there is no further mention or discussion of this. Calibration appears to rely in some way on a different c-ToF, but exactly how that is being used is not clear. It sounds like not all compounds have authentic/synthesized standards, and if not, how are retention times determined? Estimating collection and transfer efficiency by comparing GC-CIMS signal to CIMS signal likely suffers from an assumption of equal transmission of all isomers, but it is not clear if that is being accounted for because there is little discussion of how it is being applied.

Although details of instrument calibration were presented in the supplement, but we agree that the main text lacked necessary detail. We have rewritten the calibration section, pulling information from the supplement to discuss instrument sensitivity and better clarify how peak assignment and retention times were determined. Issues regarding the transmission of individual isomers is now discussed in more detail in the discussion section. Additional information of how this was addressed is provided below.

3) Demonstrating the capabilities by field deployment is a valuable addition, but the authors seem to be focusing more on the actual isoprene oxidation chemistry and science than in thinking about this as a proof-of-concept. Several detailed chemical questions regarding unimolecular reactions, etc., are addressed and discussed all within a few short paragraphs, which does relatively little to advance the instrumentation aspect of the paper, but is also too detailed and dense to give a good discussion of the scientific advances. My strong recommendation is focus on this section only as a proof-of-concept. For instance, a major advantage of this technique is the time-resolved dataset the authors show no time series of concentrations or isomer ratios, or demonstrations of the continuous operation other than inferring it from the diurnal patterns.

We have rewritten this section to focus more on the instrument data as a proof-of-concept. Although we still use isoprene oxidation products to demonstrate the capabilities of this instrument, we only briefly touch on the chemistry to describe the relevance of these compounds in the atmosphere and highlight conclusions we can draw with this isomer-specific data set. We have also updated Fig. 9 to includes a time series of the four ISOPOOH/IEPOX isomers, rather than just the diurnal averages. This demonstrates the impact of continuous operation during a portion of the PROPHET campaign.

Technical Comments:

Page 2 line 3: "can also lead to a scenario" sounds a bit odd to me, maybe just "Chemical oxidation can cause OVOCs to increase....".

The wording has been changed to "Chemical oxidation can also cause OVOCs to increase..."

Page 2 line 5: Use "In addition" or "also", but probably not both

We have removed "also" from the sentence.

Page 2 line 27: The GC paragraph sounds like it is about GC in general, in which case it would be a much larger discussion. I think the authors are mostly discussing field-deployabe/in-situ GCs here, which should be made clear.

This paragraph has been reworded to ensure that the reader knows we are discussing field-deployable GCs and not GCs in general.

Page 2 line 30-31: Though I realize it is not possible to make this list of GC-based OVOC measurements comprehensive, there are a few absences that stand out. I would include some of Allen Goldstein's measurements in this list, perhaps Millet et al., JGR 2005 which saw MVK and methacrolein as well as other OVOCs, and arguably SVTAG as I believe it can see many oxygenated gases (Zhao et al., AS&T 2013). I would also include the NOAA GC, such as Goldan et al., JGR 2004, and/or some of Jessica Gilman's work.

We have added the following citations: Millet et al., JGR 2005, Zhao et al. AS&T 2013, Goldan et al. JGR 2004 and Lerner et al. AMT 2017

Page 3 line 22: I don't understand how the air is subsampled. Is there a sample loop or something? Oh, on the head of the column - so is there a valve in the CIMS interface? There needs to be some more clarity on how sampling happens

As mentioned in the response to the first comment, we have updated Fig.1 to reflect the different instrument flow paths and have reworded this section to add more clarity to how GC sampling occurs. The air is pulled through a column (which is pre-cooled to -20 C) and analytes are trapped at the column head. Valves downstream of the column determines whether the sample gas is pulled to a scroll pump or into the CIMS (flow tube or ion source).

Page 3 line 17: What is "Low pressure" about it? It seems like a regular GC approach to me: trapping cold, then heating and pushing through N2 to a vacuum detector. I

gather the "LP" aspect is the large bore column, which reduces carrier gas pressures? A short mention or discussion of this would be helpful.

Thank you for this comment. We now highlight the pressure at which the GC operates (noting the location of this pressure measurement in Fig. 1) which is less than 260 mbar during compound separation. We have also provided an additional paragraph (page 5, lines 26) in order to provide more details about the benefits of this technique. It reads: ". . .low pressures support the use of short, large bore columns without significant loss in peak separation. This becomes especially advantageous during cryotrapping as this large I.D. column allows for a greater volume of analytes to be pulled through and trapped, beneficially impacting the instrument signal to noise. In addition, low pressure conditions also allow for faster analysis times and lower elution temperatures (Table 2). The decrease in analysis time provides this instrument with sufficient time resolution to capture diurnal variations in measured species (one GC cycle per hour), while lower elution temperatures allow this method to be used on thermally-labile species, extending the range of compounds that can be analyzed."

Page 3 line 28: The compounds aren't separated by the temperature controller, they are separated by the GC, using a ramp controlled by the controller.

We have changed the wording of this sentence.

Page 3 line 29: "each plate" is not that clear. Do the author's mean "on either side of the GC manifold/housing"?

We have rewritten this section such that this wording is no longer used. We have also updated Fig. 2 to show the approximate locations of the heaters on the GC assembly making it clear that the heaters are located on the outside surfaces of the GC manifold.

Page 3 line 30: What is the purpose of the flow tube? It makes a big difference later, but it's not clear what the function is. I would think interaction with ions, but seems to happen latter.

As mentioned above, we have added additional details on page 4 lines 1-3. The purpose of the flow tube is to ensure the analyte stream is well mixed prior to chemical ionization by CF3O-.

Page 6 line 16-21: FT and HS is asymmetric naming, with one after the approach and the other after the result. FT and IS (ion source) would be preferred,

We now use this recommended naming scheme.

Page 6 line 23: It's not clear why introduction at the source causes longer interaction times. Does fragmentation affect/complicate calibration?

Introduction at the source allows analytes to interact as soon as CF3O- forms, rather than later downstream (as in the case of FT introduction). The interaction time between the analytes and the ions increases by approximately 10 fold. This information has been added to this section.

Fragmentation does occur and can cause some discrepancies between the GC signal and the direct CIMS sampling, as discussed in the main text. However, by comparing chromatograms obtained by sending the flow to the IS vs FT, we can see that the concentrations determined by these two methods are comparable once the enhancement factor is accounted for. We do note that we see more fragmentation ions when operating in IS mode, but this is likely due to the higher signal to noise, which would allow us to observe them unlike when performing GCs through the FT. The section now includes this information.

Page 7 line 14: When the authors say "directed" do they mean direct sampling, or analysis by GC? Given the fragmentation in the HS method, it seems to me that the latter is necessary.

The compounds are sampled into the flow tube directly (without passing through the GC). The passage has been reworded to make this clear.

Page 7 line 15: Which standards are available/synthesized and which are not? Later,

for instance in Figures 7, 10, and 11, the author's seem to know the elution orders of many specific isomers - are these all from authentic or synthesized standards?

Authentic standards were used to determine instrument sensitivity, while a combination of synthesized standards (which varied in purity) and chamber experiments were used to determine GC elution order. The section has been rewritten to provide a better understanding of how elution order was determined. Assignment of many of the isoprene products is described in Teng et al., 2017.

Page 7 line 16-20: It is not clear what the purpose of the c-ToF-CIMS is, or where it is discussed or first mentioned. Is it that there are known sensitivities with that instrument, so that is the purpose of the average sensitivity difference? How does it help to compare to the c-ToF? This paragraph needs generally more explanation to be made clearer.

We have extensive calibration of the HRToF-CIMS using four gas standards (HCN, SO2, hydroxyacetone and glycoaldehyde). These calibration gases are simultaneously sampled on the cToF which uses the same ion chemistry as the HRToF-CIMS and for which we know the sensitivities of many other compounds. Because we observed that the cToF-CIMS was 1.4x more sensitive than the HRToF-CIMS for the standard gases used, we applied this factor across all analytes to estimate the HRToF sensitivities to the compounds discussed in this study. As mentioned above, this information was originally available in the Supplement but has been moved to the main manuscript to reduce confusion.

Page 8 line 27: for which species are losses observed?

We have observed poor transmission of IEPOX. We make mention of this and clarify how we can determine its transmission when ISOPOOH (which has higher transmission) is present.

Page 9 line 2-3: This is an interesting approach that potentially provides very nice confirmation of compounds for which standards aren't available. However, do the authors know that all isomers are transferred equivalently, and that all isomers can be seen? If total isomer-resolved signal is less than direct CIMS signal, that could be due to incomplete transmission of all isomers as seems to be implied by the author's, but it could also be due to complete transmission of one isomer but not the others, or complete transmission of all observed isomers but the presence of non- or poorly-observed other isomers. It is not clear to me that this approach fully works, and no effort is made to validate it here.

We have provided proof-of-concept of using this approach in the field using IHN which has high transmission through the GC. We also agree that poor transmission can be due to a single isomer rather than a sum of all isomers. This has been observed to be the case of the two isomers of IEPOX (GC transmission about 67%) compared to ISOPOOH (GC transmission $\sim$ 100%). We now clarify how we can use this method in laboratory experiments to distinguish between which isomer(s) are responsible for compound with poor transmission, without the use of synthesized standards.

Page 9 Section 3.2: Dilution will solve the humidity problem, but only reduces and does not solve the problem of reactions on the adsorbent from ozone or other oxidants. To collect these compounds, an ozone scrubber is probably out of the question, but have the author's done any tests to evaluate the impact of ozone on these compounds under typical sampling conditions?

In response to this comment, we have performed additional experiments in which we trapped ozone from an air sample containing 200 ppb ozone on the column. In this experiment, we oxidize isoprene under high NOx conditions to produce IHN. We use IHN because its reaction rates with ozone are isomer-specific. In addition, during this oxidation approximately 100 ppb of NO2 was produced, providing another oxidant to test. There was no evidence that either oxidant affected the IHN, even at higher dilutions (15x) and colder trapping temperatures (-50C). This information has been added in the discussion section.

Page 10, Section 4: How long were these campaigns and/or measurement periods?

We have now included dates to specify when and for how long for each campaign took place.

Page 10, line 18: "Isomers" is a more common term than "mass analogous", or the authors could use "isobaric", the mass spectrometric term for having the same mass

We have changed "mass analogues" to "isobaric,"

Figure 3d-f: Both axes are CIMS signals, but one is labeled normalized IHN signal, and the other as CIMS signal. Why not label the right as m/z 104 signal, or water signal or something comparable?

We have changed the axes labels of Fig 3 to reflect these changes.

Figure 8: The period of GC elution (black line) seems to have significantly lower scatter, even during background periods - is that real, and if so why is that?

Signal scatter is lower during the GC elution. We proposed an explanation for that in the caption stating: "Changes in the amount of flow entering the ion source during direct CIMS and GC-CIMS sampling directly correlate with the signal to noise seen during each operating mode. The increased flow rate through the ion source during the GC sampling mode results in higher ion counts and increased signal to noise."

---

## Author Comment (AC2) · 31 Oct 2018

Thank you for the carful read of our manuscript. We have addressed the comments and modified our manuscript accordingly.

Instrument description section:

Suggest devoting significantly more time on the first paragraph describing the instrument and Figure 1. Even if the details of this are in previous papers. The title speaks of low pressure chromatography but the words "low pressure" are mentioned only two

times and with very little description of it, how it works, what pressures the GC operates under etc.

We agree and the following changes have been made in response to this comment:

* Fig. 1 has been updated and now differentiates different instrument flow paths including direct CIMS sampling, GC trapping and GC elution.

* The first paragraph of this section has been expanded and now briefly contrasts laboratory studies described in previous papers with this automated GC-CIMS design.

* We have included an additional paragraph under the GC subsection to further discuss the low pressure chromatography. This includes listing the pressure that the GC operates (< 260 mbar depending on if passing the GC output through the ion source or flow tube) at as well as the benefits that result from operating under these conditions. This passage reads: "As mentioned above, connecting the GC outlet directly to the mass spectrometer allows the entire column to remain at sub-ambient pressures during elution... low pressures support the use of short, large bore columns without significant loss in peak separation. This becomes especially advantageous during cryotrapping as this large I.D. column allows for a greater volume of analytes to be pulled through and trapped, beneficially impacting the instrument signal to noise. In addition, low pressure conditions also allow for faster analysis times and lower elution temperatures (Table 2). The decrease in analysis time provides this instrument with sufficient time resolution to capture diurnal variations in measured species (one GC cycle per hour), while lower elution temperatures allow this method to be used on thermally-labile species, extending the range of compounds that can be analyzed."

It is not clear at all how the cryofocusing is accomplished. Please clarify this section and take the time and space to describe the different parts of Figure 1 – particularly the cryofocus and low pressure aspects of the GC. The very high flow rates are an interesting aspect of the design and this should be highlighted and explained.

We have taken better care to better describe how cryofocusing is accomplished while referencing several components labeled in Fig. 1. Also, as mentioned above, further discussing the low pressure aspect of this GC also highlights the high flow rates of this instrument.

Since this paper is about the description of an automated field-hardened instrument, provide more details on how the various components of the instrument are fitted together and how the automation was accomplished. The description of the instrument is not concise and does not have a good flow.

More detail has been provided, particularly in the GC section. This section has also been rearranged to highlight key design components followed by a concise description of sample collection and elution operating parameters.

Often very indirect language is used which results in the manuscript being too wordy perhaps at the expense of not providing concise details.

An example: 5/11: "During the collection of analytes on the head of the column, it is important that the temperature remains stable, as sizable fluctuations in temperature adversely affects the chromatography. To control the trapping set point. . ."

Could be replaced by something like: A PID control loop using heaters and the resistance temperature detector (RTD, F3102, Omega) located on the GC column ring (Fig. 2, #2 on diagram) were used to maintain fine control over the temperature set points during cryofocusing. This is needed to obtain reproducible chromatography.

Suggest going through the Instrument description sections and make clear declarative statements where possible and appropriate of the instrument design. Provide details needed for the reader to grasp the primary design features and justification for them without having to refer to previous papers.

The specific example and other instances in the instrument description now use more direct language. Primary design features and justification for their use are now described in more detail in both the instrument description and discussion sections

5/12: To control the trapping set point, we utilize the heaters and the resistance temperature detector (RTD, F3102, Omega) located on the GC column ring (Fig. 2, #2 on diagram)

Perhaps show the heaters on the diagram

We have updated Figure 2 and heater locations were added to the diagram.

5/14: In addition, during trapping we only use the solenoid valve connected to the 0.15 mm I.D. restrictor as this valve provides a $CO_2$ flow that is adequate to maintain the GC temperature ($\sim$10 slm)

?????

We apologize for the confusion. We have reworded the passage to clarify the purpose of the different $CO_2$ valves.

Calibrations and backgrounds:

7/15: However, as standards are not available for many species mentioned in this work, these calibration experiments were simultaneously performed on the c-ToF-CIMS to directly compare the compound sensitivities between these two instruments. On average, the c-ToF-CIMS was 1.4 times more sensitive. . .

I know what you mean here and it is explained further in the supplement but please rewrite more clearly in the main section as other readers will not get this on a quick read through.

We have rewritten this passage and incorporated some information that was provided in the supplement to make the calibration procedure clearer to the reader.

7/21: We use two methods to quantify the instrumental background signals caused by interfering ions present at targeted analyte masses. In the first method, the instrument

undergoes a "dry zero" where the CIMS flow tube is overfilled with dry nitrogen so that no ambient air is sampled during this time. In this method, the humidity within the instrument changes substantially compared with ambient measurements. The second method passes. . ..

How do the two methods compare?

The following text has been added to the section as a response to this comment: "The dry zero is most similar to the GC measurements and can assess the health of the instrument over the course of a campaign (i.e. these backgrounds should not change over time), while the ambient zero captures background signals that are adjusted for the water dependent sensitivity of the compounds measured during direct CIMS sampling."

Discussion

8/16: The largest technical challenge in developing a field-deployable GC was the design of a sampling system capable of collecting and separating compounds with minimal analyte degradation.

Why is this true for a field-deployable system? Seems that you need those same characteristics for a laboratory based system. The difference in a field – deployable system one would think is in getting the sample undisturbed to the instrument which is not addressed – and possibly trivial if the right sampling manifold is used (also not discussed). Referring to my opening comments, the question here is whether more of the details of the system – or a similar prototype system are discussed in previous papers. I suggest that these details be repeated here for the reader. Address what was specifically done in the field-deployable GC versus the prototype laboratory system.

We agree. The goal of this section was to discuss the difficulty of minimizing losses while transmitting reactive compounds through this GC system, rather than the difficulty of constructing a field-deployable GC as a whole. As such we've rearranged this section to better reflect this. In addition, parallels between this GC system and the

laboratory prototype have been added to the Instrument Description section. This is necessary to highlight the automated nature of this instrument.

Field Performance:

10/9: "However, instrument upgrades performed prior to the Caltech study were able to greatly reduce GC downtime and significantly improved the chromatography, despite other operating conditions remaining mostly unchanged."

This in a nutshell exemplifies the main problem with the paper. What were the instrument upgrades? Isn't this what the paper is supposed to be about?

We have removed this passage from this section and it is now incorporated in Section 2 when we discuss features of the GC design. Additional information about these upgrades are also provided in the Supplement.

Figures:

Fig 1. Enlarge the LP-GC portion of the drawing with better detail on the valving and cryofocusing aspects

The authors have updated Fig. 1 as described in a previous comment. We have also included better detail on some of the valves (e.g. the 4-port valve at the head of the column)

Fig 2. Enlarge drawing and add heaters on solenoid positions

Fig 2. has been updated to include more information, including heater position on the assembly.

Small thing. . .

4/22: For the studies detailed in this paper. . .unnecessary to start the sentence with this. Check paper for other such incidences

This text was removed and other instances in this paper were corrected as well.